# Correctional Work: Reflections Regarding Suicide

**DOI:** 10.3390/ijerph18084280

**Published:** 2021-04-17

**Authors:** Christine Genest, Rosemary Ricciardelli, R. Nicholas Carleton

**Affiliations:** 1Faculté des Sciences Infirmières, Université de Montréal, Montreal, QC H3C 3J7, Canada; 2Centre d’étude sur le Trauma du Centre de Recherche de l’Institut Universitaire en Santé Mentale de Montréal, Montreal, QC H1N 3V2, Canada; 3Centre for Research and Intervention on Suicide, Ethical Issues and End-of-Life Practices, Montreal, QC H3C 3P8, Canada; 4Department of Sociology, Memorial University of Newfoundland, St. John’s, NL A1C 5S7, Canada; rricciardell@mun.ca; 5Department of Psychology, University of Regina, Regina, SK S4S 0A2, Canada; nick.carleton@uregina.ca

**Keywords:** suicide prevention, public safety personnel, correctional workers, qualitative research, occupational mental health

## Abstract

The Public Health Agency of Canada declared suicide a public health problem in Canada (2016). Employees working in correctional services, researchers find, experience high rates of life-time suicidal ideation in comparison to other public safety professionals and the general population. Suicide behaviours (i.e., ideation, planning, attempts, death) are a multifactorial phenomenon, explained in part by the Interpersonal-Psychological Theory of Suicide that suggests attempted suicide is facilitated by perceived burdensomeness, a lost sense of belonging, a feeling of hopelessness, and a progressively reduced fear of death, as well as capacity and planning to engage a lethal attempt. In the current study, we unpack the factors that can influence suicide behaviours as reported by correctional workers. Our intent is to make explicit the experiences of a small sample (*n* = 25) of correctional workers in relation to suicidal behaviours, highlighting stories of recovery and acknowledging the importance of facilitating psychologically safe workplaces. Analysis entailed an inductive semi-grounded emergent theme approach. Participants identified certain risk factors as being able to induce suicidal ideation, such as marital or family problems as well as difficulties at work (i.e., bullying or difficult working conditions). Having children and a partner may act as factors preventing suicide for those with ideation. Participants sought help from professionals, such as their family doctor, a psychologist, or the Employee Assistance Program (EAP); however, the lack of perceived organisational supports and recognition of the issue of suicide by the employer are two elements that can hinder the search for help.

## 1. Introduction


*“I believe it’s unreasonable to think that anyone hasn’t thought about it or known people that have committed suicide”.*
*(participant 410)* 

Death by suicide among Canadians occurs at an annual rate of approximately 11 people per 100,000 [1], or about 4000 deaths each year [2], making suicide the ninth leading cause of death in Canada [3]. Approximately one-third of deaths by suicide occur among persons aged 45–59, with men being three times more likely to die by suicide in comparison to women [2]. The Public Health Agency of Canada [2] reports that 11.8% of all Canadians have had suicidal thoughts in their lifetime, 4.0% have made a plan, and 3.1% have attempted death by suicide in their lifetime. In 2016, the Public Health Agency of Canada [4] published a federal framework for suicide prevention describing suicide as a public health problem requiring further research to develop better solutions.

Researchers have recently turned their focus to suicidal behaviours among public safety personnel (PSP, e.g., border services officers, correctional workers, firefighters, paramedics, police, public safety communicators, and search and rescue personnel) [5]. A systematic review evidenced firefighters, paramedics, and police as being at an elevated risk for suicidal thoughts and behaviours relative to the general public [6]; however, the review was not focused on Canada and did not include other PSP because of insufficient published data. Subsequent research with a large and diverse sample of Canadian PSP found evidence of the past-year and lifetime suicidal thoughts (i.e., 10.1%; 27.8%), planning (i.e., 4.1%; 13.3%), and attempts (i.e., 0.3%; 4.6%) at rates higher that than rates found among the general population [7]. Correctional workers (CWs) reported some of the highest rates of past-year and lifetime suicidal thoughts (i.e., 11.0%; 35.2%), planning (i.e., 4.8%; 20.1%), and attempts (i.e., 0.4%; 8.1%), which underscored the need to better understand their experiences. Moreover, in the same dataset, 54.6% of CWs screened positive for one or more mental disorders (e.g., major depressive disorder, posttraumatic stress disorder (PTSD), generalised anxiety disorder, social anxiety disorder, panic disorder, alcohol use disorder) [8].

The challenges with mental disorder symptoms among CWs can be expected to explain, at least in part, the prevalence of suicidal behaviours [7,9]; in addition, the risk and resiliency factors associated with suicidal behaviours in the general population (e.g., marital status, sex) [9] also appear to be factors for PSP [7,10,11]. PSP, including CWs, appear to experience more exposures to potentially psychologically traumatic events (PPTE) [5] than the general population [12], which can be expected to exacerbate mental health difficulties and suicidal behaviours [9,12,13]. The CW work environments include unique challenges that may further increase mental health risks [14]. For example, those working in institutional correctional services may face challenges tied to overcrowding, violence, infectious disease, and contraband, to name only a few factors that make the prison work environment precarious and uncertain [15,16,17].

Suicidal behaviours (i.e., ideation, planning, attempts) involve multifactorial phenomena [18,19,20]. Specific precipitating factors for a suicide attempt can include a romantic break-up, bereavement, job loss, or bullying [21]. Other risk factors include unemployment, being a man, social isolation, and low socioeconomic status, which increase vulnerability to death by suicide [22]. There are also protective factors, such as being married, having social support, and willingness to accept help [23,24]. The Interpersonal-Psychological Theory of Suicide is one theory that can be readily employed to describe the suicidal behaviours of PSP [20,25,26]. The model suggests suicide attempts are facilitated by perceived burdensomeness, a lost sense of belonging, a feeling of hopelessness, and a progressively reduced fear of death, as well as the capacity for a lethal attempt. CWs perform invisible labour at work and report often feeling unsupported by management, which may perpetuate a lost sense of belonging. The CW work environment (e.g., dealing with distressed prisoners/parolees/probationers, monitoring suicidal behaviours in others) involves increased exposures to PPTE, which may inform CW well-being. The work of CWs may also contribute to acquired capability to die by suicide, which is a core concept of the Interpersonal-Psychological Theory of Suicide [20,25].

### Current Study

Drawing from the same dataset as the current study, Carleton and colleagues found that provincial CWs further evidenced high levels of mental disorders, with 58.2% screening positive for one or more mental disorders (i.e., major depressive disorder, PTSD, generalised anxiety disorder) [27]. In addition, CWs further reported high levels of past-year and lifetime suicidal thoughts (i.e., 7.0%; 26.6%), planning (i.e., 2.6%; 11.9%), and attempts (i.e., n/a; 5.2%) [28]. The previous research results with CW data have focused on quantifying self-reports regarding suicidal behaviours, risk factors, and resilience factors; however, very little work has focused on the contextualised experiences of CWs regarding suicide, resulting in a lacuna of knowledge and scholarship internationally. The current study uses qualitative data provided by a large Ontario sample of CWs employed either in community, institutional, or administrative correctional services who participated in a survey regarding mental health. The current study is designed to make explicit the experiences of CWs in relation to suicidal behaviours, highlighting stories of recovery and acknowledging the importance of facilitating psychologically safe workplaces for everyone [29].

## 2. Methods

The current study data were drawn from a larger research project examining CW well-being. The design details are presented elsewhere [27,30,31]; in short, cross-sectional data were collected via an online survey available from 8 December 2017 to 30 June 2018 based on established guidelines for web-based surveys [32]. CWs in the Ontario provincial correctional system were invited by email to answer socio-demographic questions, respond to self-report scales assessing the presence of different symptoms, and provide commentary in response to open-ended items. Open-ended items often followed a scale and provided space for respondents to give context and detail about their experiences. In the current article, we analyse the comments of CWs who, after responding to scales assessing CW lifetime and past-year experiences with suicide, were asked to provide “additional comments”. The question was intentionally broad to provide a space for CWs to voice any interpretations, concerns, or ideas related to suicide.

CW participants were recruited by email, using both the organisational listserv and that of their union, and sent by a representative of the Ministry of the Solicitor General and/or the Ontario Public Service Employee Union (OPSEU). The email described the study purpose, how to participate, and details regarding confidentiality, anonymity, and informed consent. The email content underscored participation as completely voluntary and provided a link to the survey which was hosted on the platform Qualtrics. Participants could stop the survey and continue answering questions at a later time without starting over. The time required to complete the entire survey varied from between 25 and 40 min.

A total of 25 participants answered the open-ended question about the experience of suicidal ideation and suicidal behaviour. Responses from the 25 participants were used for the present study (see Table 1). The current sample was small; as such, we removed participant demographics from our results to protect participant privacy. Our analysis entailed an inductive thematic analysis process [33]; more specifically, a constructed semi-grounded emergent theme approach [34,35,36], where emergent themes were derived from the data (i.e., discussion that was consistent across respondents) that involved a multistep coding process utilising QRS NVivo Pro. The data were first coded by a graduate-level research assistant, and then recoded for reliability and validity by the second author. The emergent themes were then organised and explained, centralising the voices of respondents. Unlike traditional grounded theory, a theory did not emerge from the data. Accordingly, our study is semi-grounded, where themes emerged organically, but we did not disengage from our theoretical knowledge to create a new theoretical position [34].

## 3. Results

The results are structured such that we begin by unpacking the death by suicide-related experiences of participants. We discuss the different ways death by suicide can present within the CW population. Next, we turn to the risk factors underpinning movement toward suicide behaviours to shed light on why some CWs reach positions of detailed ideation, planning, or attempts. Prior to reporting help-seeking behaviours employed by CWs to assist with suicidal behaviours, we unpack the protective factors that participating CWs report prevented them from dying by suicide.

### 3.1. Presentation of Suicide Ideation among CWs

Participants had much to report regarding their additional comments after completing the quantitative assessment of their lifetime and past-year experiences with suicidal behaviours. Many participants simply wrote that they had “thoughts about it” (participant 349), with some participants reporting a belief that everyone has likely had such thoughts or at minimum does know someone that has experienced death by suicide. Participants differentiated between having thoughts about suicide and actually wanting to die by suicide. For example, participant 659:
There is a difference between wanting the suffering to end, wanting to be still and wanting to be dead, although the nuance of each is a very thin line. I would say I’m passively suicidal; not actively doing anything to make it happen but also not afraid of death if it were to come.

Participant 659′s words suggest a desire to end suffering but a want to live. They explain that “the nuance of each [wanting to end suffering versus want to die] is a very thin line”. The participant further explains how “thin” the line is in acknowledging being “passively suicidal”—welcoming death or not actively trying to stay alive, but not actively trying to die. Such a position is concerning for many reasons, including that attitudes toward death could impact propensities to engage in risky decision-making or behaviours [37,38].

Other participants discussed their own experiences contemplating and engaging in suicide behaviours more actively. For example, participant 675, explained that his loved ones learned of his ideation after finding a note he had written to accompany his plan. Nevertheless, he felt able to hide this suicide risk from those close to him, despite the evidence. The desire to die was apparent per the plan and note, and the efforts to hide their suicidality underscored wilful isolation in suffering. CW training regarding suicide prevention and monitoring, or institutional culture engagements with mental health, may also encourage, even inadvertently, CWs to hide their own personal struggles under a professional visage [15,39]. The CW job often entails presentations of stoicism and toughness [39]. Their words reveal isolation and complexities surrounding death by suicide. Other participants reported finding solace and support from other people, like participant 740, who reported that, “I had a close friend that helped me out of this state”. Another participant presented the physical effects of the ideation, planning, and attempt to die by suicide as impactful, even debilitating; specifically, participant 1047, wrote “my neck and back were so tense it caused an ongoing migraine”. The participant’s words describe physical effects of the psychological state, including consequences for their physical well-being. Recognising that people may be more likely to seek professional intervention for physical conditions, physical symptoms, like participant 1047′s migraine, could lead them to seek medical intervention without disclosing their associated psychological difficulties. The participants underscore the diverse ways suicidal behaviours can be experienced, with some suffering in isolation, others reaching out for support, some with physical ramifications, and others passively open to death.

### 3.2. Reported Risk Factors for Suicidal Behaviours

Respondents discussed diverse areas that ‘triggered’ thoughts regarding suicide; specifically, family-related experiences, work experiences, or experiences from their youth. Participants described marital challenges as generating suicide ideation. For example, participant 161, wrote that their “thoughts were from after a separation, only lasted a few days then thoughts went away”, whereas participant 1092 described that his thoughts emerged after “Marital breakup”. Both participants reported temporary experiences with suicidal ideation associated with changes in marital status and associated hardship. Participant 41 described challenges within her family that motivated suicidal thoughts; specifically, a “traumatic event in my family left me feeling hopeless and lost”. The participant described her family experiencing a PPTE that was difficult to navigate and produced feelings of hopelessness, collectively compromising her well-being, leading to suicide contemplation. Participant 692 described similar challenges: “First kid, wife was experiencing violent mood swings, kid wasn’t sleeping, I still had to work full time”. The participant speaks to how balancing work and home life, when challenging situations (e.g., birth of a child and wife’s violent mood swings) complicate home life, increased his mental health vulnerability, which contributed to his suicidal ideations.

Some participants considered their employment or occupational experiences as risk factors for their suicidal thoughts. Here, participants spoke to either conditions of the workplace such as “all due to poor working conditions and poor working environment” (participant 546) and the experiences of workplace harassment, “My employer bullying me and or targeting me drives me to think of suicide” (participant 1037). In the first excerpt, the participant speaks to negative overall workplace conditions as having a detrimental effect on their well-being and eventually leading to suicide contemplation. In the second excerpt, the participant speaks to the negative impact of workplace bullying and the role of the employer and sometimes colleagues as a “driving” factor toward suicide ideation. Another participant described a prior unemployment experience when he “had no job. Life without money…is very difficult” (participant 1058) as a factor influencing his suicidal behaviours. The participants appear to have described situations where they felt burdened (e.g., not having money to take care of themselves) and experienced thwarted belongingness (e.g., being bullied by co-workers, being unemployed). The situations appear stable and unchanging, which could lead to a sense of hopelessness and therein the suicidal ideation.

Participants also described experiences from their youth as being part of their suicidal history which could eventually influence future suicidal behaviours, which is entirely consistent with previous PSP research [13]. Participant 360 wrote that, “I was young and feeling overwhelmed by life. Thought about honestly doing it. But I loved and continue to love life”. The participant describes genuine suicidal ideation in their youth due to being “overwhelmed by life”. Other participants described youthful suicidal ideation as occurring when they were “young [and] dumb, thought about it, never tried” (participant 471) or attributed suicidal ideation to “middle school hormones” and “bullying” (participant 609, both quotes). The participants describe those difficult youth experiences as influencing contemporary suicidal behaviours, but the negative impact of “bullying” or being “overwhelmed by life” appeared as consistent themes that shape experiences with suicide.

### 3.3. Reported Resilience Factors for Suicidal Behaviours

Participants described different factors that affected their decision not to attempt to die by suicide. Participants described personal intrinsic factors and external factors, including having family and pets. Intrinsic factors took diverse forms. For example, a participant (800) wrote that, “Emotional collateral damage/consequences. Amount of effort needed” were factors that prevented their attempting to die by suicide. Specifically, the participant did not attempt suicide because of the effort required (i.e., suicide was not perceived as easy or simple) and the consequences for others. The participant implies difficulties with the time and energy required to put affairs in order, plan, and attempt to die. Other participants described their “fear” or their “willpower” as preventing their attempting to die by suicide. Regarding the former, participant 455 wrote “fear of suffering”, whereas participant 647, wrote that “willpower” prevented their own attempt to die by suicide. Collectively, the participants describe an internal quality that saved their life—a reliance on self that stopped ideation and planning from becoming an attempt.

External factors that had a role in “saving lives” and preventing death by suicide have often been related to having children or a partner [40,41]. Participant 570 wrote “my kids”, whereas participant 1037 wrote “my wife”, identifying their relationships as critical protective factors against attempting suicide. Another participant described being the “Primary care provider to a family member” (participant 379) as the factor which kept them from attempting suicide—specifically, he did not want to harm someone particularly close to him. Another participant attributed “my animals” (participant 30) as a relational commitment mitigating suicide attempts. Overall, the participants describe critical protective factors such as relationships as having a role in preventing suicide attempts.

### 3.4. Seeking Help

Many participants who sought professional assistance for their experience with suicidal ideation explained they “now go to counselling” (participant 30). Some participants visited with a family doctor prior to beginning counselling, like participant 184, who stated, “I did go to my doctor and have received help for my thoughts and counselling to get better—this is ongoing”. Some participants reported visiting their doctor very recently for assistance, “I rarely go to the doctor, but I did see him three days ago to seek some assistance” (participant 455). Other participants were actively on leave (e.g., “off on stress leave, counselling” participant 507), or reported taking medication to support their mental health (e.g., “have been on anti-depressants for 6 years”, participant 191). The participants describe treatment as ongoing to maintain and achieve mental wellness. A negative experience with help-seeking was also shared by participant 261, who reported having “Sought psychiatric help. Treated so poorly in the community. Denied PTSD doctor. EAP (employment assistance program) interview(er) was so stunned by my 40 min diatribe. Left her speechless. Not very helpful. Made me feel worse”. The participant’s words describe a less than positive experience on multiple fronts when seeking treatment, a poor community response, and a negative experience, for him, with the employee assistance program, which left him feeling “worse”. The idea that help-seeking could result in a new kind of mental health injury is evidenced by the excerpt and suggests the importance of cultural competence for providers working with CW [42].

Some participants also voiced concerns tied to their employer having insufficient understanding of suicidal behaviours. One participant wrote that the Canadian suicide prevention website was not accessible through their work computers, which blocked easy access to find a crisis centre (participant 150). Another participant reported that “management bugged me LAST MONTH on why I do not wish to do suicide watches” (emphasis original, participant 240). The participants’ experiences suggest that personal experience with suicide can increase the challenge of performing their work and can consequently leave the employee feeling their employer is unsupportive toward employee mental health.

Some participants reported that they stopped seeing a mental health professional despite perceiving the professional as beneficial. The reasons were often tied to limited sessions (e.g., “I only got three sessions”, participant 735) or challenges with scheduling (participant 669). Other participants reported having found support elsewhere and thus stopped seeing the professional (e.g., “spoke to my fiancée”, participant 1092). Still other participants reported collaboratively deciding with their mental health professional that they no longer required ongoing professional help (participant 768). Reasons participants discontinued treatment are relatively congruent and largely focused on scheduling, cost, and limited availability due to employment restrictions.

## 4. Conclusions

The current study results provide details regarding CW experiences with suicidal ideation and planning. CW frequently experience PPTE including violence and death [12], which may potentiate a loss of their sense of fear in the face of death and pain. The loss of fear associated with death is an element of Joiner’s interpersonal theory of suicide [21], such that when fear of dying reduces, capacity for suicide can increase [20,43]. Suicidal behaviours require attention and must be taken seriously. Challenging behaviours associated with suicidality can include increased risk-taking; for example, CWs may have a lowered fear of dying that might facilitate CWs engagement with risky and potentially fatal behaviours at work or in their day-to-day lives.

The participating CWs reported several potential risk factors for suicidal behaviours (e.g., familial or relational problems, including divorce, separation, family conflict, bullying) similar to those reported by the general population and by other PSP. Participants reported the main cause for suicidal behaviours is relationship difficulties in their personal or professional life. Conversely, having reliable loved ones can act as protective factors for suicidal behaviours, which is consistent with the interpersonal theory of suicide and the impact of thwarted belongingness [20,21,22]. Some CWs also mentioned feeling a sense of hopelessness in the face of the situation. Feelings of hopelessness are an important element of suicidal behaviours [44]. CWs in the current study described important interactions between their work context and the personal life that make disentangling personal and occupational risk factors difficult, which is consistent with previous reports [11]. The findings suggest that the most effective interventions might require a multitude of changes to personal and occupational factors.

The current results indicate that CWs may face barriers related to admitting difficulties with suicidal behaviours and accepting help; for example, a participant reporting that despite writing a suicide “note” that was found, he denied his compromised mental health and convinced family and friends that he was healthy. Such activities underscore the ongoing challenges CWs face regarding mental health stigma [14]. CWs may actively avoid talking about their mental health needs, limiting opportunities for getting help [45,46], but there was evidence that some participants engaged in reporting and help-seeking despite the stigma [14]. Greater reductions in mental health stigma among CWs may be extremely beneficial to support their mental healthcare and might start with more frequent and explicit discussions about suicidal behaviours in the workplace.

CWs described their family physician as a primary contact for help-seeking, which is consistent with previous research results from among other populations, such as PSP and the general public [47]. Our current preliminary results with CWs are consistent with previous research on PSP [48,49], in that symptoms of psychological distress can present with physical symptoms (e.g., migraines). Future research on PSP who present with physical symptoms may provide important insights into mental health disorders and suicidal behaviours. Such information could help inform family physicians and other health professionals/care providers about the importance of assessing for comorbid mental health disorders. Early recognition can support early intervention, which may help to mitigate the elevated risk for suicidal behaviours.

Some of our participating CWs reported feeling less understood or even misunderstood by the mental health professionals who offer help, which is consistent with previous research among police [11,49,50] and other PSP [47]. Participating CWs appeared to understand that their work environment (e.g., prison, community, supporting prisoners/probationers/parolees), organisational culture, and frequent PPTE exposures (e.g., violence, abuse, suicidal behaviours of others) underscore the importance of cultural competence for perceiving psychological services as potentially effective [42]. CWs in Ontario, Canada, have readily accessible employment assistant programs, where providers could demonstrate such competencies as part of reducing barriers to help-seeking.

Participants cited relational commitment to family, peers, and friends as resilience factors. Relational commitment may facilitate a sense of belongingness [20,25,26]. Peer support programs are often premised on peers as helpful for detecting psychological distress and behavioural changes in their colleagues as a function of relational commitment. PSP often turn to peers, friends, and family for help [47]; therefore, training certain peers to properly detect symptoms of psychological distress and provide appropriate supports and referrals may be a way forward, encouraging prevention and intervention [49]. Peer support programs may also offer collegial support as a function of shared experiences among CW [42,51]. Offering diverse resources could help overcome the constraints reported by CWs as barriers to mental healthcare (e.g., lack of accessibility, costs). The current results highlight challenges faced by CWs with respect to suicidal behaviours and provide a preliminary understanding of their experiences.

Few scholars have studied suicide within the scope of correctional work. Our study is preliminary, given our small cross-sectional sample and lack of ability to probe participants for more information about their responses. Researchers are also influenced by preconceptions that influence data analyses. The interpersonal theory of suicide may have influenced our understanding of suicide among CWs. Continued research in the area is warranted to validate how CWs interpret the influence of risk and protective factors in the development of their suicidal behaviour. Further research should include longitudinal qualitative interviews with CWs about their mental health, experiences with suicide, and the influencing factors and proactive strategies, as well as diagnostic longitudinal interviews, all to improve our understanding of how CWs health changes with occupational tenure. We recommend longitudinal research that follows the course of CWs, starting at recruitment and into their occupational tenure, to better understand the factors underpinning suicidal behaviours across time, space, and place.

## Figures and Tables

**Table 1 ijerph-18-04280-t001:** Participants’ description.

By Gender	
Gender	Number of participants
Female	7
Male	17
Non-binary/Non-conforming	1
Grand Total	25
By Organisational Level and Occupation	
Organisational Level	
Occupation	
Administrative (Community)	0
Administrative Support	
Administrative (Institutional)	3
Administrative Assistant	
Clinical/Nursing	
Correctional Officer	1
Other Support	2
Operational (Community)	1
Probation and Parole Officer	1
Operational (Institutional)	20
Clinical/Nursing	1
Correctional Officer	19
Institutional Management	
Organisation Not Specified	1
Grand Total	25

## Data Availability

The data that support the findings of this study are available from the corresponding author upon reasonable request.

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
