# Peer review of "Correctional Work: Reflections Regarding Suicide"

_ijerph, 2021, doi:10.3390/ijerph18084280_

Round 1

Reviewer 1 Report

Thank you for the opportunity to contribute to your work.

Here are some thoughts I had, while reading your article. There is an undeniable necessity to adress distress and suicidal ideations in CW and other PSP. 

It takes time before we are introduced to the experience of correctional officers in term of suicidal behaviour. In the abstract and the introduction, it would be nice to start with correctional officers more directly, telling us more straight forwardly why we need to adress suicidal risk in these CW.

Why concentrate on the interpersonal theory of suicide in the introduction, rather than any other? We understand in the discussion that you found support for the acquired capability component in your data, but there are other models that can help you. Why this one?

Why is it relevant to concentrate specifically on CW suicide risk? Are there factors in their work that can increase risk? The fact that they work with distressed people, have to monitor suicide behaviour in others, do an unpopular job, etc are relevant elements to note here

P2, l97 : what do you mean by “Canadian provincial sample”? is it pan Canadian?

I would specify in the introduction of the “current study” section that it is secondary data analysis.

P3, L106 : I am confused about the range of CW involved. It says here that only Ontario based CW participated. It would be important to specify more clearly

P3, L124 : I would like to have information on the overall sample and information specifying the small subsample of 25 who answered the open ended question.

P3, L127 : a reader  not familiar with constructed semi-grounded emergent theme approach needs more detailed information on the method used to analyse the data.

Also did you analyse these contents in regards with other results from the survey (symptoms, experiences….). And what is the sample description details?

P3, L137 : you mention PSP here, but the study is about CW only. Why not specify here?

P4, L162 : you say that work environment could help them hide their suicidality. How?

P5, L 205 : this refers more to the entrapment model of suicide than the interpersonal model. What do you think?

Participants 360, 471 and 609 seem to describe past suicide ideations, not current.

P6, l260 : it is a strong indication that we have to be very careful to provide positive experience with mental health care, to avoid increasing distress

P7.l310 : discussion on suicidal behaviour =  where? can work settings be an adequate place to do information and screening?

P7, l315 : is that a specificity of CW and PSP? Do you have hypotheses why?

P7, l323 : why are PSP less understood by mental health professionals? Less than the general population?

P7, l328 : in order to recommend work based intervention, you should make your case stronger about the impact of work on suicide risk for CW

It would be very interesting to analyse participants’ experience with suicide before and after they started working as CW. How their work experience as a CW changed their risk? How do we understand that CW have a higher rate of distress than others?

When you cite participants, it would be nice to have their age and gender. Since this may be a very masculine work environment, it would be interesting to identify gender variations.

You may gain some space by not rephrasing participants’ words (ie: participant 41, page 4)

I am all for doing secondary analyses. They can bring depths and details to otherwise short-handed and dry statistical studies. In this case, I would like for the authors to explain better what is the added value of a small sample analysis like this one, when compared with the quantitative large scale studies. Also, it would be important to have some information on the results from the larger quantitative study from which this data is extracted. Who are those 25 respondents who took the time to express themselves about their suicidal thoughts and behaviours?

Overall, it seems that there are quite a few typos and grammatical errors left.

Author Response

1. It takes time before we are introduced to the experience of correctional officers in term of suicidal behaviour. In the abstract and the introduction, it would be nice to start with correctional officers more directly, telling us more straight forwardly why we need to address suicidal risk in these CW.

Response: We revised the abstract and introduction, such that we address CW suicide ideation and suicide attempts earlier. In light of the reorganization with data on suicide behaviours of PSP and CW, we moved down the paragraph on the Interpersonal theory of suicide for flow.

2. Why concentrate on the interpersonal theory of suicide in the introduction, rather than any other? We understand in the discussion that you found support for the acquired capability component in your data, but there are other models that can help you. Why this one?

Response: In the PSP literature, we find the interpersonal theory of suicide is most appropriate for explaining their suicide behaviours, due to the work environment and the exposure of CWs to potentially psychologically traumatic events. The theory is well suited to explain why CWs are at risk of having suicide ideation and eventually attempt suicide because of the acquire capability. We have added a sentence mentioning that there are a number of theories to explain suicide behaviours, however scholars have tended apply the Interpersonal theory of suicide to PSP suicide. We continue in this manner by applying the theory to CWs in specific.

3. Why is it relevant to concentrate specifically on CW suicide risk? Are there factors in their work that can increase risk? The fact that they work with distressed people, have to monitor suicide behaviour in others, do an unpopular job, etc are relevant elements to note here

Response: We added some explanation on the work-related risk factors that are specific to CW. In the CW environment, the fact that CWs are performing invisible labour and often feeling unsupported by management may perpetuate a lost sense of belonging. Moreover, the CW work environment (e.g. dealing with distressed prisoner/parolees/probationers, monitoring suicide behaviours in others), with the increased exposure to potentially psychologically traumatic events may contribute to the acquire capability that is a core concept of the theory.

4. P2, l97 : what do you mean by “Canadian provincial sample”? is it pan Canadian?

Response: We have revised to clarify that the sample is from Ontario and is not pan-Canadian.

5. I would specify in the introduction of the “current study” section that it is secondary data analysis.

Response: It is primary data analysis as the author also collected the data.

6. P3, L106 : I am confused about the range of CW involved. It says here that only Ontario based CW participated. It would be important to specify more clearly

Response: We have clarified, stating it is from Ontario, and breaking down who are included in the participants.

7. P3, L124 : I would like to have information on the overall sample and information specifying the small subsample of 25 who answered the open ended question.

Response: We now provide a table detailing sample characteristics.

8. P3, L127 : a reader  not familiar with constructed semi-grounded emergent theme approach needs more detailed information on the method used to analyse the data.

Response: We now explain the semi-grounded approach applied to data collection.

9. Also did you analyse these contents in regards with other results from the survey (symptoms, experiences….). And what is the sample description details?

Response: We have provided more information about the sample characteristics and prior analysis involving the same dataset.

10. P3, L137 : you mention PSP here, but the study is about CW only. Why not specify here?

Response: We have removed peripheral discourse regarding PSP and more clearly focused on CWs. We still draw from PSP literature as there is more information that is relevant in that literature.

11. P4, L162 : you say that work environment could help them hide their suicidality. How?

Response: We revised the sentence to explain that the suicide prevention training and the organisational culture toward mental health and suicide may encourage CWs to hide their own struggles under a professional visage.

12. P5, L 205 : this refers more to the entrapment model of suicide than the interpersonal model. What do you think?

Response: Although the statement could be linked to the entrapment model of suicide, we also feel it fits with the interpersonal model of suicide. Specifically the hypothesis that, “The simultaneous presence of thwarted belongingness and perceived burdensomeness, when perceived as stable and unchanging (i.e., hopelessness regarding these states), is a proximal and sufficient cause of active suicidal desire” (Van Orden & al., 2010).

13. Participants 360, 471 and 609 seem to describe past suicide ideations, not current.

Response: The answers of those CWs was linked to lifetime or past year suicide ideation. We modified the beginning of the paragraph to clarify that the data focuses on past experience that may still influence current suicidal behaviours.

14. P6, l260 : it is a strong indication that we have to be very careful to provide positive experience with mental health care, to avoid increasing distress

Response: Thank you, we added your suggestion to the paragraph.

15. l310 : discussion on suicidal behaviour =  where? can work settings be an adequate place to do information and screening?

Response: We added a sentence in the paragraph to promote open discussion about suicide and mental health in the work environment to help reduce the stigma.

16. P7, l315 : is that a specificity of CW and PSP? Do you have hypotheses why?

Response: Most CW and PSP are male, which may interact with the presentation of physical symptoms and help-seeking. Our current study is qualitative and grounded; as such, we do not include hypotheses.

17. P7, l323 : why are PSP less understood by mental health professionals? Less than the general population?

Response: PSP are less understood because of their work environments, which includes extremely frequent exposures to potentially psychologically traumatic events. Indeed, many CWs reported feeling misunderstood by mental health professionals. Participating CWs asked to be treated by professionals who understand their work environment or work experience, so the CWs do not have to explain the nuance. We added a sentence to explain the phenomenon.

18. P7, l328 : in order to recommend work based intervention, you should make your case stronger about the impact of work on suicide risk for CW

Response: We added information on the work environment and how it could impact suicide risk (in the introduction and in the discussion) as a function of CWs feeling insufficiently well understood by mental health professionals.

19. It would be very interesting to analyse participants’ experience with suicide before and after they started working as CW. How their work experience as a CW changed their risk? How do we understand that CW have a higher rate of distress than others?

Response: We have added this to the study limitations and suggested future work should include a longitudinal study design.

20. When you cite participants, it would be nice to have their age and gender. Since this may be a very masculine work environment, it would be interesting to identify gender variations.

Response: We were asked to remove such possibly identifying information by the Ministry and have added a sentence to explain that decision.

21. You may gain some space by not rephrasing participants’ words (ie: participant 41, page 4)

Response: The rephrasing was done in an effort to clarify our interpretations of participant responses, which makes us hesitant to remove the additional content. Recognizing the trade-offs between manuscript length and clarity we are willing to make additional revisions pending an editorial request.

22. I am all for doing secondary analyses. They can bring depths and details to otherwise short-handed and dry statistical studies. In this case, I would like for the authors to explain better what is the added value of a small sample analysis like this one, when compared with the quantitative large scale studies. Also, it would be important to have some information on the results from the larger quantitative study from which this data is extracted. Who are those 25 respondents who took the time to express themselves about their suicidal thoughts and behaviours?

Response: The article is not based on secondary analysis and we provide more information about the sample.

23. Overall, it seems that there are quite a few typos and grammatical errors left.

Response: We have thoroughly edited the manuscript.

Reviewer 2 Report

The manuscript addresses a relevant issue in suicide behaviors, the suicidal behaviors in personnel working in penitentiary institutions from a qualitative perspective.

The introduction has relevant and pertinent information to be able to understand the magnitude and theoretical framework to address the issue.

Nevertheless, there are some aspects in the manuscript that need to be improved.

The aim of the study is not clearly stated the Current study section in the introduction.

It is necessary to improve the description of the methodology. Please, mention the number of total participants in the survey. Mention the selection criteria of the 25 participants. Were there inclusion or exclusion criteria to the answers selected? for instance, length.

Describe in detail the process of coding and generating topics, for example, were there independent judges in the coding? how were discrepancies between judges resolved? Please, describe the steps in the coding process.

The Reported resilience factors for suicidal behaviours are especially important, nevertheless, there were sparsely discussed.

Few typos are in the manuscript.

Author Response

1. The aim of the study is not clearly stated the Current study section in the introduction.

Response: We modified the last sentence of the “current study” section, to clarify and provide more information.

2. It is necessary to improve the description of the methodology. Please, mention the number of total participants in the survey. Mention the selection criteria of the 25 participants. Were there inclusion or exclusion criteria to the answers selected? for instance, length.

Response: We have added more methodological information, including about the participants.

3. Describe in detail the process of coding and generating topics, for example, were there independent judges in the coding? how were discrepancies between judges resolved? Please, describe the steps in the coding process.

Response: We now clearly explain the steps in the coding process and our analysis.

4. The Reported resilience factors for suicidal behaviours are especially important, nevertheless, there were sparsely discussed.

Response: We added some information on the importance of relational commitment in terms of resilience factors, as well as how relational commitment is linked to the sense of belongingness and supports the implementation of peer support programs.

5. Few typos are in the manuscript.

Response: We have carefully edited the manuscript.

Reviewer 3 Report

In this article, Genest and colleagues reported on the experiences of a sample of correctional workers in relation to suicidal behavior, highlighting stories of recovery and acknowledging the need of facilitating psychologically safe workplaces.

The authors surely explored an important and delicate topic, as correctional workers are one of the most at-risk public safety personnel populations.

The study was built upon a well justified and referenced rationale. The introduction is well written and provides clear information to understand the topic and the purpose of the research. The context is appropriately described, which is important to allow the reader to make judgments of whether the results might be transferable to another setting. An explanation of why and how the methodological approach was used is present. Results presentation is accurate, and the Discussion is properly refined from both a conceptual standpoint and in terms of readability, as it is the manuscript as a whole.

I have no major concerns about the quality of the present submission.

Nonetheless, some minor issues should be fixed in order to further improve the quality of the work:

  1. A reflection on how the researchers’ position, preconceptions, and biases could have influenced the findings and their interpretation should be added, as these aspects are significant in qualitative research.
  2. Besides correctly stating that the study is preliminary (LL 338-339), a few words should be spent on its possible limitations.
  3. The last sentence of the Methods section (LL 124-128) should be moved to the Results section, according to the STROBE statement for observational studies (https://www.strobe-statement.org/index.php?id=available-checklists).
  4. The type of study design (cross-sectional) should be declared at some point of the manuscript.

Author Response

1. A reflection on how the researchers’ position, preconceptions, and biases could have influenced the findings and their interpretation should be added, as these aspects are significant in qualitative research.

Response: We have added a sentence at the end of the discussion to detail the possible influence. Researchers are also influenced by preconceptions that can influence data analysis; indeed, the interpersonal theory of suicide which was our conceptual framework have influenced our understanding of suicide among CWs. Continued research in the area is warranted to further validate with CWs regarding how their interpretations interact with risk and protective factors in the development of their suicidal behavior.

2. Besides correctly stating that the study is preliminary (LL 338-339), a few words should be spent on its possible limitations.

Response: We have added more information on study limitations mainly on the impact of our

preconception on the research project

3. The last sentence of the Methods section (LL 124-128) should be moved to the Results section, according to the STROBE statement for observational studies (https://www.strobe-statement.org/index.php?id=available-checklists).

Response: We have revised the methods.

4. The type of study design (cross-sectional) should be declared at some point of the manuscript

Response: We state the study is cross-sectional.

Round 2

Reviewer 2 Report

The authors made all the requested modifications. Best regards.